# Sleep Deprivation Increases Facial Skin Yellowness

**DOI:** 10.3390/jcm12020615

**Published:** 2023-01-12

**Authors:** Akira Matsubara, Gang Deng, Lili Gong, Eileen Chew, Masutaka Furue, Ying Xu, Bin Fang, Tomohiro Hakozaki

**Affiliations:** 1Procter & Gamble Innovation G.K., 7-1-18 Onoedori, Chuo-ku, Kobe 651-0088, Japan; 2Procter & Gamble International Operations SA SG Branch, 70 Biopolis Street, Singapore 138547, Singapore; 3Procter & Gamble Technology (Beijing) Co., Ltd., 35 Yu’an Rd, Shun Yi Qu, Beijing 101318, China; 4Department of Dermatology, Kyushu University, 3-1-1 Maidashi, Higashi-ku, Fukuoka 812-8582, Japan; 5The Procter & Gamble Company, Mason Business Center, 8700 Mason Montgomery Road, Mason, OH 45040, USA

**Keywords:** sleep deprivation, skin yellowness, skin redness, skin tone, dull skin appearance, bilirubin, carotenoids, IL-6, biopyrrin

## Abstract

Sleep shortage is a major concern in modern life and induces various psycho-physical disorders, including skin problems. In cosmeceutics, females are aware that sleep deprivation worsens their facial skin tone. Here, we measured the effects of sleep deprivation on facial skin yellowness and examined yellow chromophores, such as bilirubin and carotenoids, in blood serum as potential causes of yellowness. Total sleep deprivation (0 h sleep overnight, N = 28) and repeated partial sleep deprivation (4 h sleep for 5 consecutive days, N = 10) induced significant increases in facial skin yellowness. The higher yellowness was sustained even after both sleep deprivation types stopped. However, circulating levels of yellow chromophores were unchanged in the total sleep deprivation study. Neither circulating interleukin-6 nor urinary biopyrrin levels were affected by total sleep deprivation, suggesting that apparent oxidative stress in the body was not detected in the present total deprivation protocol. Facial redness was affected by neither total nor repeated partial sleep deprivation. Therefore, blood circulation may play a limited role in elevated yellowness. In conclusion, facial skin yellowness was indeed increased by sleep deprivation in our clinical studies. Local in situ skin-derived factors, rather than systemic chromophore change, may contribute to the sleep deprivation-induced elevation of facial skin yellowness.

## 1. Introduction

Sleep plays a critical role in maintaining physical and mental wellbeing and health [1,2,3]. Sleep assists the body’s homeostatic systems and maintains or repairs various biological functions [4]. Although the question of why we need to sleep has not been completely resolved, many deprivation studies have revealed three critical functions of sleep [5]: (i) restoring energy supplies of the body, (ii) anti-inflammation as a defense reaction, and (iii) restoration of neuronal synaptic homeostasis that maintains learning and memory functions. These are all crucial for the maintenance of life, influencing not only mental and physical health but also probably the homeostatic condition of the skin, the largest organ of the body.

Owing to changes in lifestyle, sleep duration has been declining for the last three decades, irrespective of ethnic background [6,7]. Shortage of sleep is a common feature in the modern lifestyle worldwide. The use of smartphones may also impact the duration and quality of sleep [8], as does stress from work and study [9,10]. Sleep duration is known to differ between the sexes, with females tending to go to bed and fall asleep earlier than males; meanwhile, females wake more often during the night and more commonly suffer from a lack of sleep than males [11]. In the cosmeceutical field, females are aware that sleep deprivation can cause various skin problems and worsen their facial skin appearance.

In accordance with this, acute sleep deprivation has been shown to reduce perceived health and attractiveness [12]. In addition, Kim et al. reported that skin hydration, skin barrier function, and skin elasticity are decreased by sleep deprivation [13]. Jang et al. also reported the impact on skin condition of repeated sleep deprivation, at 4 h of sleep per day for 6 nights. They found that this sleep deprivation decreased skin hydration while worsening skin desquamation, transparency, elasticity, and wrinkles [14]. Oyetakin-White et al. also studied the difference in skin aging between good sleepers and poor sleepers using the Pittsburgh Sleep Quality Index questionnaire. They concluded that good sleepers had a lower aging score [15].

In terms of skin tone, many females claim that sleep deprivation induces dullness of facial skin. However, few studies examining the effects of sleep deprivation on facial skin dullness have been carried out. Sallow and/or dull skin appearance is greatly attributable to the yellow components of skin tone [16]. Dullness and yellowness of the skin have been reported to be major concerns of Asian women [17]. To understand the association between skin dullness and sleep deprivation, we conducted two independent clinical trials: total sleep deprivation (0 h sleep overnight) and repeated partial sleep deprivation (4 h sleep for 5 consecutive nights). We found that both sleep deprivation protocols significantly aggravated facial skin yellowness with reciprocal reduction of facial skin hydration. No significant changes in circulating levels of yellow chromophores such as bilirubin and carotenoids were detected. These results suggest that the sleep deprivation-induced elevation of facial skin yellowness is probably regulated by in situ skin condition rather than the circulating chromophores.

## 2. Materials and Methods

The total sleep deprivation study was conducted in Xi’An, China, from October to November 2016. The repeated partial sleep deprivation study was conducted in Beijing, China, in November 2018. Prior to the execution of each study and data collection, institutional review board approval of the study protocols was obtained from the Ethics Committee of Beijing HaiTai HeChuang Technology Co., Ltd. (IRB number: BJIC016-027) for the total sleep deprivation study, and from the Ethics Committee of Cosmetics Technology Center, Chinese Academy of Inspection and Quarantine (IRB number: 2018-008-DY-100) for the repeated partial sleep deprivation study. All participants provided written informed consent. The subjects underwent health checks periodically on site.

### 2.1. Subjects

Subjects in the total sleep deprivation study comprised 28 Chinese females aged 22 to 30 (27.0 ± 2.2, mean ± standard deviation) years old. Meanwhile, subjects in the repeated partial sleep deprivation study comprised 10 Chinese females aged 24 to 32 (28.3 ± 2.2) years old. They self-reported being in general good health and having no serious medical history or sleep-related issues. They provided written informed consent and their participation in the study was on a completely voluntary basis.

### 2.2. Study Design

In the total sleep deprivation study, the pre-deprivation phase was from Day −3 to Day 0. Total sleep deprivation (0 h sleep overnight) was performed on the night of Day 0. All subjects (N = 28) were followed without restriction of sleep time from Day 1 to Day 7 in the post-deprivation phase. To keep all subjects awake overnight, they stayed at the measurement facility. Some activities such as group chat and light indoor exercise (e.g., walking or stretching) were organized periodically. The subjects were allowed to bring their own electronic devices (e.g., smartphones, tablets, and laptops) or items such as cards to entertain themselves. They were also allowed to consume water and light snacks, but not caffeine.

We next conducted a preliminary study to elucidate the effects of repeated partial sleep deprivation. In the repeated partial sleep deprivation study, the pre-deprivation phase was set as Day 0. All subjects (N = 10) were allowed to sleep only 4 h per night for 5 consecutive nights from Day 1 to Day 5, and were followed without restriction of sleep time from Day 6 to Day 11. The partial sleep deprivation was conducted at each subject’s home. The study investigator instructed the subjects to go to bed at 12 a.m. and wake up at 4 a.m. To control the time of falling asleep, they were given tasks (e.g., writing a short essay) and were requested to send a social networking service (SNS) message indicating the completion of this task to the study investigator immediately before 12 a.m. To manage the time of waking up, they were requested to respond to a SNS message from the investigator that was sent immediately after 4 a.m.

In both studies, the duration of sleep was measured using a wrist-worn Actigraph GT9X three-axis accelerometer (Actigraph, LLC, Pensacola, FL, USA). This device records high-resolution raw acceleration data, which are converted into a sleep measure using publicly available algorithms developed and widely used by members of the academic sleep and activity research community [18,19].

### 2.3. Measurement of Skin Yellowness, Redness, and Hydration

The measurement of facial skin tone and hydration was conducted between 8 a.m. and 10 a.m. each day in both studies. The facial images were captured using a VISIA-CR4.3^®^ imaging system (Canfield Scientific, Parsippany, NJ, USA). The facial images were recorded in RAW format (3744 × 5616 pixels) and converted into BMP format to perform image analysis. The facial skin yellowness and redness were calculated using a computerized image analysis algorithm, as has been described previously [20,21]. The measurement of facial skin hydration was conducted on the cheek area using Corneometer CM825 (Courage + Khazaka, Cologne, Germany).

### 2.4. Blood and Urine Sample Collection

In the total sleep deprivation study, blood (10 mL) and urine (30 mL) were also collected between 8 and 10 a.m. each day. The collected samples were stored in a freezer at −80 °C until measurement. Total blood bilirubin (sum of direct and indirect bilirubin) was quantified using Human Total Bilirubin (Vanadate Oxidation Method) Kit (Beijing Leadman Biochemistry Co., Ltd., Beijing, China). Total blood carotenoids, including lutein, zeaxanthin, cryptoxanthin, α-carotene, β-carotene, and lycopene, were quantified using high-performance liquid chromatography (HPLC) at Hope Testing Service (Tianjin, China). Blood interleukin-6 (IL-6) and urinary biopyrrin levels were also quantified at Hope Testing Service, using Human IL-6 High Sensitivity ELISA BMS213HS (eBioscience Inc., San Diego, CA, USA) and Human Bilirubin Oxidation Metabolite (Biopyrrin) Kit (Shanghai Enzyme-Linked Biotechnology Co., Ltd., Shanghai, China), respectively.

### 2.5. HPLC Analysis

Reverse-phase, gradient HPLC system was used for separation of carotenoids in human blood serum. The HPLC system consisted of a Waters 2695 HPLC system (Waters Corporation, Milford, MA, USA) with Waters 996 PDA detector and a prontosil C30 column (250 × 4.6 mm, 5 μm particle size) from Bischoff Chromatography (Leonberg, Germany). The flow rate, detection wavelength, and injection volume were 1.0 mL/min, 450 nm and 30 μL, respectively. The HPLC mobile phase was methanol/methyl-tert-butyl ether/water (83:15:2, *v/v/v*, with 1.5% ammonium acetate in the water; solvent A) and methanol/methyl-tert-butyl ether/water (8:90:2, *v/v/v*, with 1% ammonium acetate in the water; solvent B). The gradient procedure at a flow rate of 1 mL/min (room temperature) was as follows: (1) 100% solvent A for 1 min; (2) a 1 min linear gradient to 70% solvent A; (3) a 3 min hold at 70% solvent A; (4) a 17 min linear gradient to 45% solvent A; (5) a 1 min hold at 45% solvent A; (6) an 11 min linear gradient to 95% solvent B; (7) a 4 min hold at 95% solvent B; (8) a 2 min gradient back to 100% solvent A, and (9) a 10 min hold at 100% solvent A for equilibrium to return to the initial condition. Using this method, lutein, zeaxanthin, cryptoxanthin, α-carotene, β-carotene, and lycopene were adequately separated. Stock standard solution of 50 ppm lutein was prepared by the direct dissolution of the lutein standard (obtained from NCRM, Beijing, China) in chloroform; the working standard solution was 5.0 ppm. Stock standard solution of 50 ppm zeaxanthin was prepared by the direct dissolution of the zeaxanthin standard (obtained from NCRM, Beijing, China) in chloroform; the working standard solution was 3.0 ppm. Stock standard solution of 100 ppm cryptoxanthin was prepared by the direct dissolution of the cryptoxanthin standard (obtained from CaroteNature GmbH, Munsingen, Switzerland) in physiological saline; the working standard solution was 5.0 ppm. Stock standard solution of 0.932 ppm α-carotene was obtained from NCRM, Beijing, China; the working standard solution was 0.932 ppm. Stock standard solution of 100 ppm β-carotene was prepared by the direct dissolution of the β-carotene standard (obtained from NICPBP, Beijing, China) in physiological saline; the working standard solution was 3.0 ppm. Stock standard solution of 50 ppm lycopene was prepared by the direct dissolution of the lycopene standard (obtained from CaroteNature GmBH, Munsingen, Switzerland) in physiological saline; the working standard solution was 5.0 ppm.

### 2.6. Statistical Analysis

All statistical analyses were performed using JMP^®^ Pro 16.1.0 (SAS Institute Inc., Cary, NC, USA). For each outcome variable, data were analyzed by fitting a linear mixed-effects model with timepoint as the fixed effect and subject as the random effect to account for within-subject variation. Timepoint effects were estimated by least squares means and compared using Student’s *t* test versus Day 0. The correlation of skin yellowing and hydration change between before and after sleep deprivation was examined using Pearson’s r. A *p*-value below 0.05 was considered to reflect statistical significance.

## 3. Results

### 3.1. Increased Facial Skin Yellowness by Total Sleep Deprivation

We first conducted the total sleep deprivation study. The facial skin yellowness values were stable and no significant change was observed in the pre-deprivation baseline phase from Day −3 to Day 0 (sleep time = 347.9 ± 89.2 min) (Figure 1).

Total sleep deprivation (sleep time = 0 min) significantly increased facial yellowness at Day 1 compared with that in the pre-deprivation phase (Day 0). Notably, even in the post-deprivation phase (sleep time = 408.3 ± 131.4 min), its elevation was sustained for at least 48 h until Day 3 and finally returned to the pre-deprivation baseline levels at Day 5 and Day 7 (Figure 1).

In contrast, facial skin hydration was transiently decreased by total sleep deprivation at Day 1 compared with that in the pre-deprivation baseline phase (Day 0). The significantly decreased skin hydration rapidly returned to the baseline levels at Day 2 through Day 7 (Figure 2).

We then examined the association between yellowness and hydration, but no significant association was detected (Figure 3).

Total sleep deprivation did not affect facial skin redness immediately and rather decreased the redness value at Day 5 and Day 7 (Figure 4).

### 3.2. Total Sleep Deprivation Did Not Affect the Circulating Levels of Bilirubin and Carotenoids

As circulating blood chromophores such as bilirubin and carotenoids may influence the yellowness of facial skin [16,22,23], we measured their blood levels. However, no significant alteration was observed in their blood levels during pre-deprivation, total sleep deprivation, and post-deprivation phases (Figure 5).

It has been reported that sleep deprivation induces oxidative stress [24] and enhances the circulating levels of interleukin-6 (IL-6) [25,26]. Therefore, we next examined the serum levels of IL-6 and urinary biopyrrin, an oxidative metabolite of bilirubin and a known biomarker of systemic oxidative stress [27]. As shown in Figure 6, neither serum IL-6 nor urinary biopyrrin level was significantly affected by total sleep deprivation.

### 3.3. Repeated Partial Sleep Deprivation Also Elevated Facial Skin Yellowness

We then conducted a preliminary study (N = 10) to address whether repeated partial sleep deprivation aggravates facial skin yellowness. Compared with findings in the pre-deprivation period (Day 0) (sleep time = 345.6 ± 140.7 min), repeated partial sleep deprivation, namely, 4 h sleep for 5 consecutive nights (sleep time = 261.4 ± 77.3 min), gradually increased facial skin yellowness at Day 1 and Day 2 (Figure 7). The elevation of yellowness became statistically significant at Day 3 and Day 5 during the partial sleep deprivation phase. After cessation of partial sleep deprivation, the higher level of yellowness was sustained and still significantly higher at Day 6. After Day 7, the yellowness gradually decreased and finally returned to the baseline level at Day 9 to Day 11 in the post-deprivation phase (sleep time = 407.6 ± 123.1 min) (Figure 7).

In contrast to facial skin yellowness, facial skin hydration was decreased by the partial sleep deprivation, which reached statistical significance at Day 5 compared with that in the pre-deprivation phase (Figure 8). The facial skin hydration rapidly returned to the baseline level in the post-deprivation period at Day 6 to Day 11 (Figure 8), as seen in the total sleep deprivation study.

No significant correlation was detected between skin hydration and yellowness in the repeated partial sleep deprivation study (Figure 9).

In accordance with the results in the total sleep deprivation study, upon repeated partial sleep deprivation, the facial redness values again fluctuated throughout the study period, without a remarkable trend of increase or decline, although there was a significant reduction at Day 10 compared with that in the pre-sleep deprivation phase (Figure 10).

## 4. Discussion

Empirical evidence has shown that sleep deprivation exacerbates dullness of the facial skin. However, few studies have examined the effects of sleep deprivation on skin tone [14,28]. To shed light on this issue, we performed two independent clinical studies, total sleep deprivation and repeated partial sleep deprivation trials, in Asian women. Notably, both sleep deprivation protocols significantly increased facial skin yellowness. In addition, the elevated yellowness was sustained even after the cessation of total sleep deprivation and repeated partial sleep deprivation.

Previous studies revealed that sleep deprivation reduced facial skin hydration [13,14]. We also observed this change upon applying both sleep deprivation protocols. The increased yellowness and decreased hydration were likely to be regulated by different mechanisms because these two parameters were not significantly associated with each other. In contrast to yellowness and hydration, facial skin redness was not affected by either sleep deprivation protocol.

Skin tone is known to be influenced by various chromophores, the most representative of which are melanin, hemoglobin, bilirubin, and carotenoids [16,22,23,29,30,31]. Among these chromophores, melanin and hemoglobin have the largest influence on overall skin tone [31]. In the present clinical studies, the duration of exposure to ultraviolet rays was not modified. In addition, skin redness was found not to be altered by sleep deprivation. Therefore, melanin and hemoglobin may play limited causative roles in the elevated yellowness.

Bilirubin and carotenoids are yellowish compounds found in the body, which can modify skin yellowness [16,22,23]. We measured the circulating levels of total bilirubin and carotenoids in the present study. We found that their blood levels were not significantly modified by total sleep deprivation. In addition, we could not detect any significant alteration in blood IL-6 levels and the oxidative stress biomarker biopyrrin. However, we could not rule out the possibility that in situ skin levels of bilirubin may be involved in the elevation of yellowness because epidermal keratinocytes are themselves capable of producing substantial amounts of bilirubin sufficient to contribute to yellowing of the skin [16].

Previous studies have assessed various effects of sleep deprivation. Kim et al. reported statistically significant changes in multiple skin biophysical parameters after one night of sleep deprivation with twenty-four healthy Korean women [13]. Sauver et al. examined the effect of acute sleep deprivation on vascular function in twelve healthy subjects [32]. Wessel et al. conducted a partial sleep deprivation study, 4 h time in bed per night for 5 nights, with 13 healthy young men and concluded that 5 nights of sleep restriction increased lymphocyte activation and the production of proinflammatory cytokines [33].

This is the first clinical study to assess the effects of sleep deprivation on facial skin yellowness. We recognize that there are several constraints in our studies. First, the number of subjects was small due to a difficulty of recruiting volunteers in the longitudinal sleep deprivation protocols. Larger scale studies might be necessary to confirm our findings. Secondly, we could not directly measure the skin levels of bilirubin and carotenoids. Further studies are warranted to elucidate the causative mechanisms of elevated yellowness induced by sleep deprivation. In conclusion, both total sleep deprivation and repeated partial sleep deprivation enhance facial skin yellowness. In addition, increased yellowness and decreased hydration are likely to be differentially regulated in sleep deprivation.

## Figures and Tables

**Figure 1 jcm-12-00615-f001:**
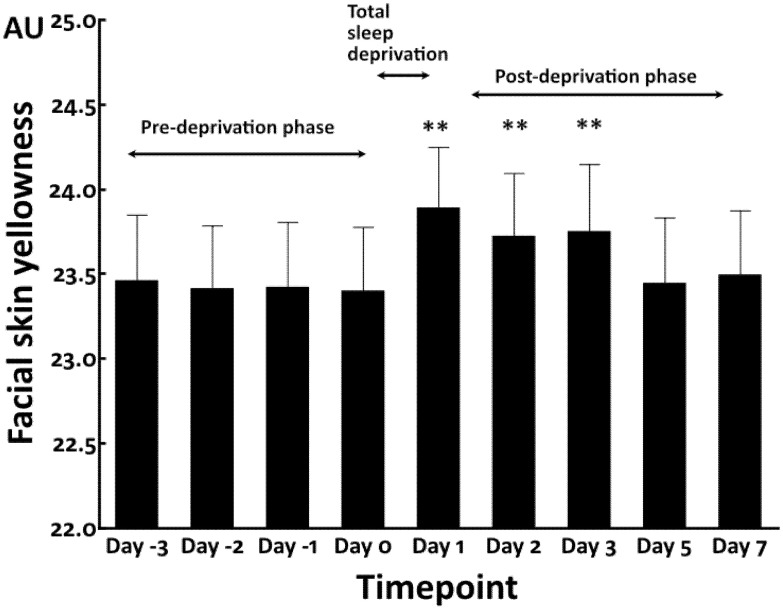
Facial skin yellowness in total sleep deprivation study. **: *p* < 0.01 compared to Day 0.

**Figure 2 jcm-12-00615-f002:**
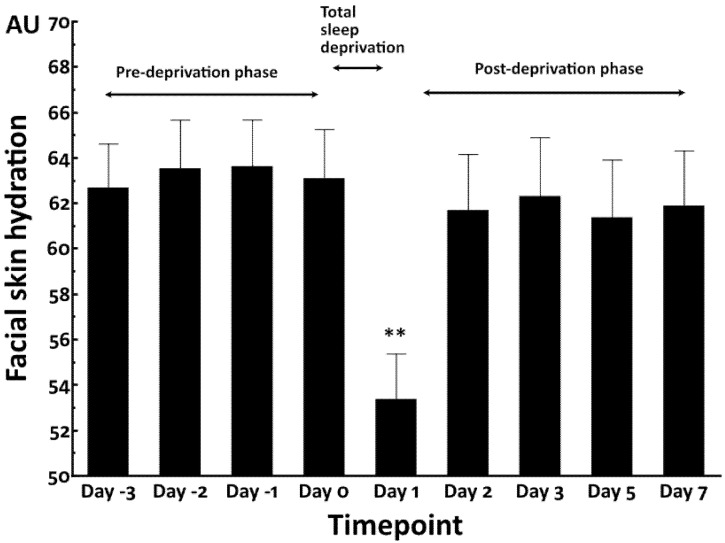
Facial skin hydration in total sleep deprivation study. **: *p* < 0.01 compared to Day 0.

**Figure 3 jcm-12-00615-f003:**
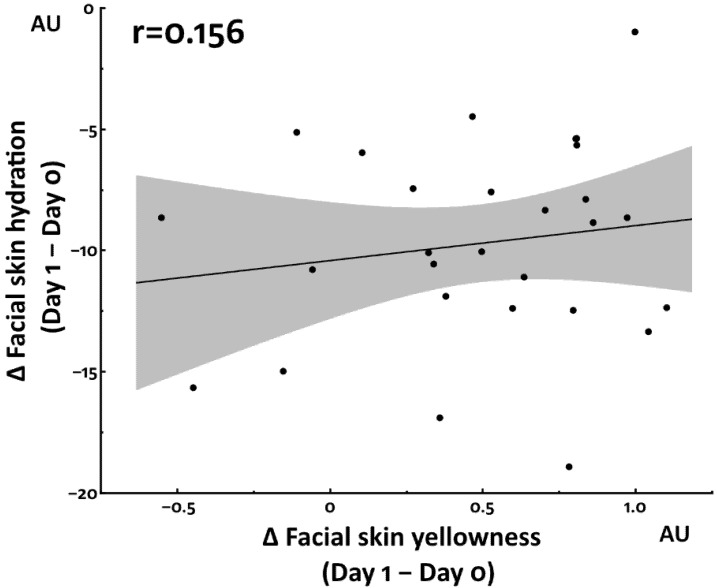
Association between facial skin yellowness and hydration in total sleep deprivation study.

**Figure 4 jcm-12-00615-f004:**
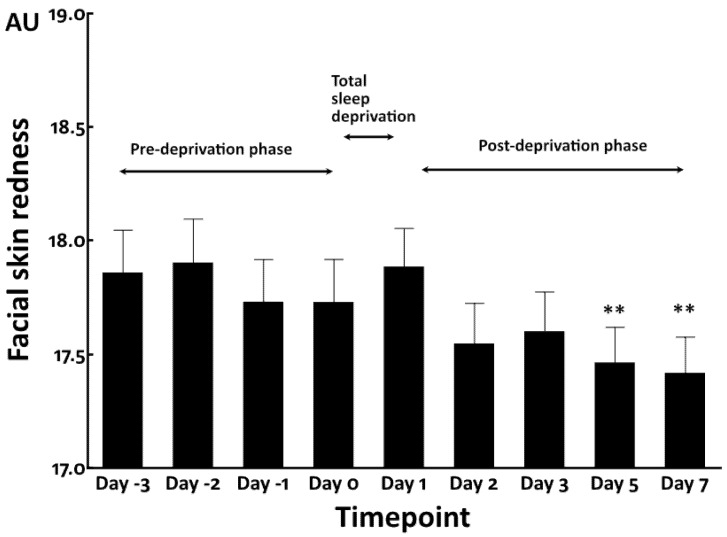
Facial skin redness in total sleep deprivation study. **: *p* < 0.01 compared to Day 0.

**Figure 5 jcm-12-00615-f005:**
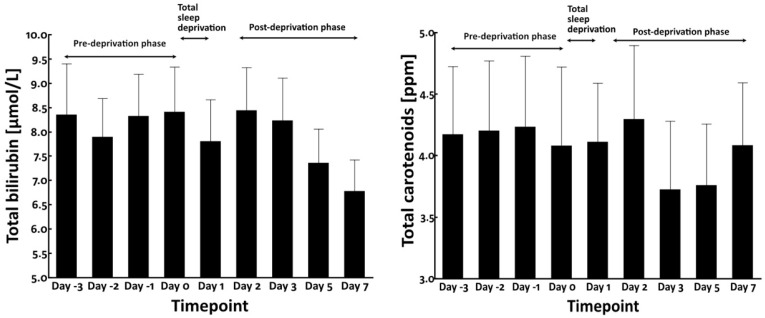
Blood levels of bilirubin (**left panel**) and carotenoids (**right panel**).

**Figure 6 jcm-12-00615-f006:**
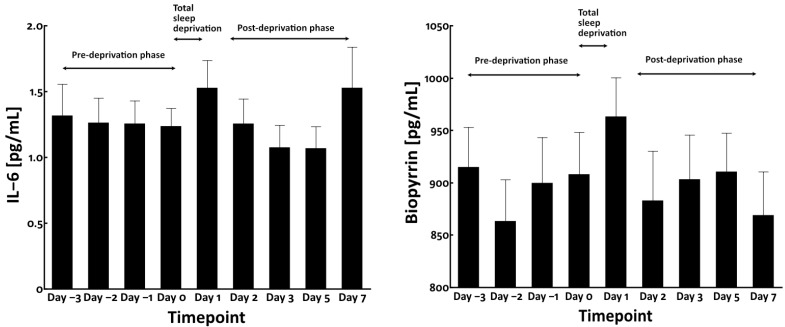
Blood levels of IL-6 (**left panel**) and urinary levels of biopyrrin (**right panel**) in total sleep deprivation study.

**Figure 7 jcm-12-00615-f007:**
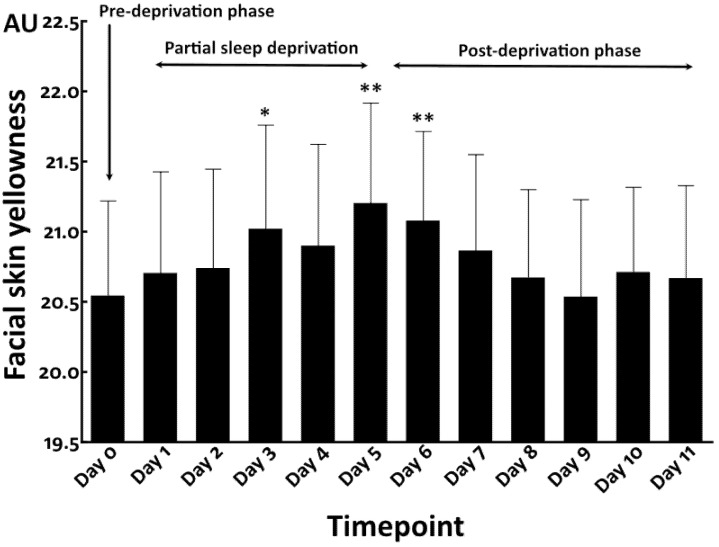
Facial skin yellowness in repeated partial sleep deprivation study. *: *p* < 0.05 compared to Day 0. **: *p* < 0.01 compared to Day 0.

**Figure 8 jcm-12-00615-f008:**
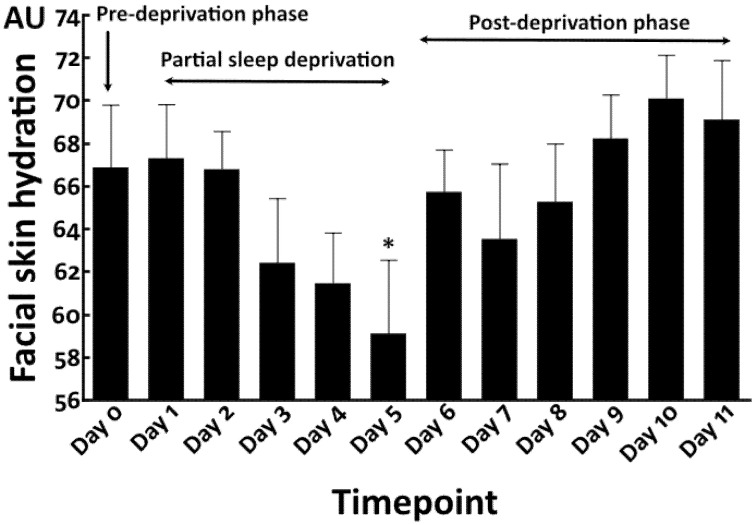
Facial skin hydration in repeated partial sleep deprivation study. *: *p* < 0.05 compared to Day 0.

**Figure 9 jcm-12-00615-f009:**
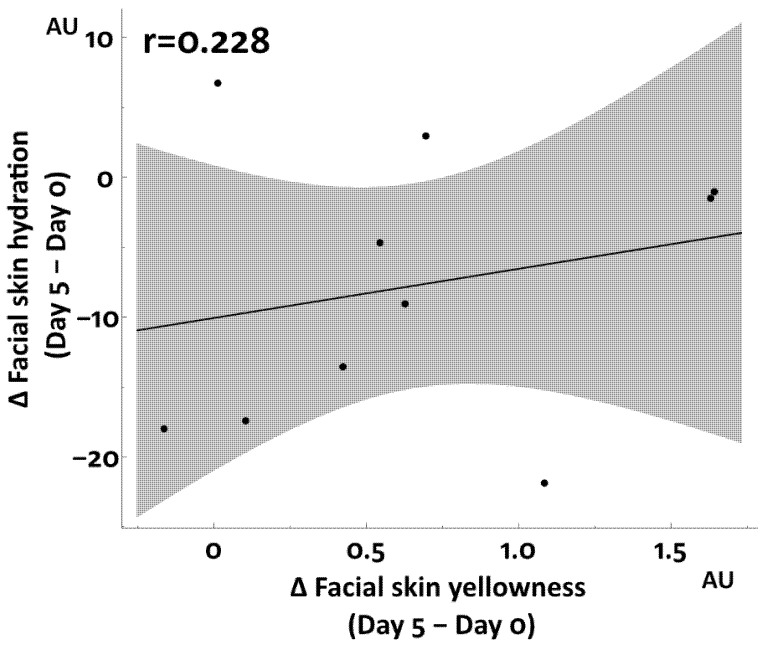
Association between facial skin yellowness and hydration in repeated partial sleep deprivation study.

**Figure 10 jcm-12-00615-f010:**
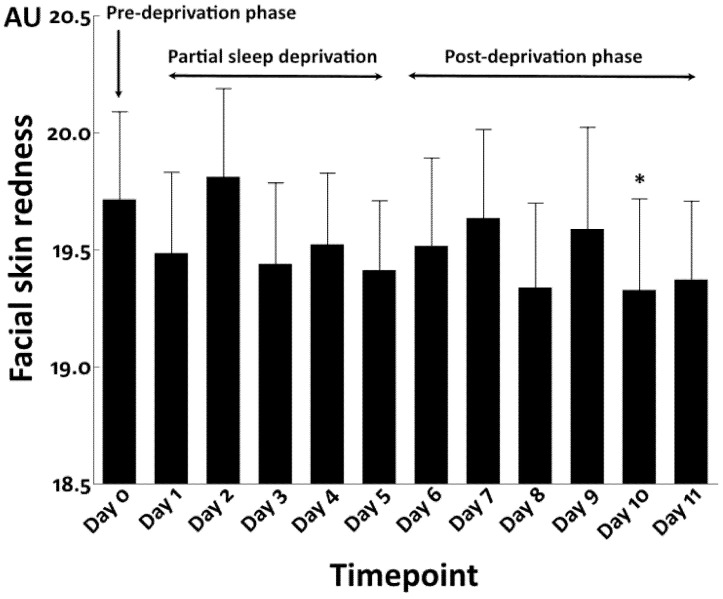
Facial skin redness in repeated partial sleep deprivation study. *: *p* < 0.05 compared to Day 0.

## Data Availability

The data presented in this study are available on request from the corresponding author. The data are not publicly available because of privacy restrictions.

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
