# Peer review of "Sleep Deprivation Increases Facial Skin Yellowness"

_jcm, 2023, doi:10.3390/jcm12020615_

Round 1
Reviewer 1 Report
The research is sound and has importance to the dermatology research area. However I have the following comments that need to be addressed.
Authors need to explain how did they generate the sample size number for the study.
Authors need to indicate the methodology of the HPLC condition (i.e. mobile phase, column, wavelength, standard curve etc) used for the quantification of blood carotenoids, including lutein, zeaxanthin, cryptoxanthin, α-carotene, β-carotene, and lycopene.
No conclusion is seen in the manuscript.
Author Response
Reply to the reviewer 1
The research is sound and has importance to the dermatology research area.
→ Thank you very much for your encouraging comment.
However, I have the following comments that need to be addressed.
Authors need to explain how did they generate the sample size number for the study.
→ We appreciate your valuable comment. The recruitment of large number of healthy females who volunteer to have a sleep deprivation over their daily norms (e.g., schooling, working) is a common challenge in controlled sleep research although we understand that larger sample size is more adequate for solid conclusion. In particular, recruiting volunteers for the longitudinal and repeated partial sleep deprivation study was higher hurdle because the subject had to compromise their normal life for 12 days. When determining the number of sample size, we referred to the prior sleep-controlled studies and adopted an equivalent number of subjects in our studies. Those authors also conducted controlled sleep studies with relatively small number of subjects. In accordance with your comment, we would like to add following underlined sentences to explain how we determined the sample size.
Line 112 to Line 123
“The recruitment of large number of healthy females who volunteer to have a sleep deprivation over their daily norms (e.g., schooling, working) is a common challenge in controlled sleep research. Kim et al. reported statistically significant changes in multiple skin biophysical parameters after one night of sleep deprivation with twenty-four healthy Korean women [13]. Sauver et al. examined the effect of acute sleep deprivation on vascular function in twelve healthy subjects [18]. Wessel et al. conducted a partial sleep deprivation study, 4 hours time in bed per night for 5 nights, with 13 healthy young men and concluded 5 nights of sleep restriction increased lymphocyte activation and the production of proinflammatory cytokines [19]. We have referred to the prior sleep deprivation protocols and recruited equivalent number of volunteers in our sleep deprivation studies.”
Authors need to indicate the methodology of the HPLC condition (i.e. mobile phase, column, wavelength, standard curve etc) used for the quantification of blood carotenoids, including lutein, zeaxanthin, cryptoxanthin, α-carotene, β-carotene, and lycopene.
→ Thank you very much for indicating the lack of the information. We added one section to the revised manuscript (2.5 HPLC analysis, Line 180 to Line 213) and described HPLC conditions.
- HPLC analysis
Reverse-phase, gradient HPLC system was used for separation of carotenoids in human blood serum. The HPLC system consisted of a Waters 2695 HPLC system (Waters Corporation, Milford, MA, USA) with Waters 996 PDA detector and a prontosil C30 column (250 x 4.6mm, 5mm particle size) from Bischoff Chromatography (Leonberg, Germany). The flow rate, detection wavelength, and injection volume were 1.0 mL/min, 450 nm and 30 mL, respectively. The HPLC mobile phase was methanol/methyl-tert-butyl ether/water (83:15:2, v/v/v, with 1.5% ammonium acetate in the water; solvent A) and methanol/methyl-tert-butyl ether/water (8:90:2, v/v/v, with 1% ammonium acetate in the water; solvent B). The gradient procedure at a flow rate of 1 mL/min (room temperature) was as follows: 1) 100% solvent A for 1 min; 2) a 1-min linear gradient to 70% solvent A; 3) a 3-min hold at 70% solvent A; 4) a 17-min linear gradient to 45% solvent A; 5) a 1-min hold at 45 % solvent A; 6) a 11-min linear gradient to 95% solvent B; 7) a 4-min hold at 95% solvent B; 8) a 2-min gradient back to 100% solvent A, and 9) a 10-min hold at 100% solvent A for equilibrium to return to the initial condition. Using this method, lutein, zeaxanthin, cryptoxanthin, α-carotene, β-carotene, and lycopene were adequately separated. Stock standard solution of 50 ppm lutein was prepared by the direct dissolution of the lutein standard (obtained from NCRM, Beijing, China) in chloroform, the working standard solution is 5.0 ppm. Stock standard solution of 50 ppm zeaxanthin was prepared by the direct dissolution of the zeaxanthin standard (obtained from NCRM, Beijing, China) in chloroform, the working standard solution is 3.0 ppm. Stock standard solution of 100 ppm cryptoxanthin was prepared by the direct dissolution of the cryptoxanthin standard (obtained from CaroteNature GmbH, Munsingen, Switzerland) in physiological saline, the working standard solution is 5.0 ppm. Stock standard solution of 0.932ppm α-carotene was obtained from NCRM, Beijing, China, the working standard solution is 0.932 ppm. Stock standard solution of 100 ppm β-carotene was prepared by the direct dissolution of the β-carotene standard (obtained from NICPBP, Beijing, China) in physiological saline, the working standard solution is 3.0 ppm. Stock standard solution of 50 ppm lycopene was prepared by the direct dissolution of the lycopene standard (obtained from CaroteNature GmBH, Munsingen, Switzerland) in physiological saline, the working standard solution is 5.0 ppm.
No conclusion is seen in the manuscript.
→ Thank you very much for your crucial comment. According to your comment, we revised our conclusion as follows:
Line 40 to 43
“In conclusion, the facial skin yellowness is indeed increased by sleep deprivation. Local in situ skin-derived factors, rather than systemic chromophore change, may contribute to the sleep deprivation-induced elevation of facial skin yellowness.”
Line 361 to 364
“In conclusion, both total sleep deprivation and repeated partial sleep deprivation enhance facial skin yellowness. In addition, increased yellowness and decreased hydration are likely to be differentially regulated in sleep deprivation.”
Thank you very much again for sharing your precious time to review this article. According to your comments, we revised the article.
We hope the revised article is now suitable for publication in JCM.

Reviewer 2 Report
The article ˝Sleep Deprivation Increases Facial Skin Yellowness˝ is an interesting article that considers the influence of sleep/lack of sleep on the level of facial skin yellowness and examines the influence of blood and urine parameters on its influence.
The study has a well-stated hypothesis, and for the most part, it is well-conducted.
The greatest advantage of this work is the attractiveness of the topic, which is mostly related to the field of aesthetic medicine. Also, it is relevant that no systemic changes were observed that could be related to increased skin yellowness.
The main drawback of the study is the uneven distribution and the small number of examinees (27 and especially 10), which make the results of the statistical analysis questionable. The repeated partial sleep deprivation study methodology should be more objective and adequately controlled. Also, the control group is not included in the trial. These deficiencies should be corrected.
The conclusion states that the increase in skin yellowness is probably a consequence of in situ events in the skin, which, however, is not supported by the results of the actual study. Although the authors stated the same as a shortcoming, it would be interesting to supplement the study with this data.
Author Response
Reply to the Reviewer 2
The article ˝Sleep Deprivation Increases Facial Skin Yellowness˝ is an interesting article that considers the influence of sleep/lack of sleep on the level of facial skin yellowness and examines the influence of blood and urine parameters on its influence. The study has a well-stated hypothesis, and for the most part, it is well-conducted. The greatest advantage of this work is the attractiveness of the topic, which is mostly related to the field of aesthetic medicine. Also, it is relevant that no systemic changes were observed that could be related to increased skin yellowness.
→ Thank you very much for your encouraging comments.
The main drawback of the study is the uneven distribution and the small number of examinees (27 and especially 10), which make the results of the statistical analysis questionable. The repeated partial sleep deprivation study methodology should be more objective and adequately controlled. Also, the control group is not included in the trial. These deficiencies should be corrected.
→ We appreciate your comment. We totally agree with the criticality of your comment. The recruitment of large number of healthy females who volunteer to have a sleep deprivation over their daily norms (e.g., schooling, working) is a common challenge in controlled sleep research although we understand that larger number of examinees is more adequate for solid conclusion. In particular, recruiting volunteers for the longitudinal and repeated partial sleep deprivation study was higher hurdle because the subject had to compromise their normal life for 12 days. When determining the number of examinees, we referred to the prior sleep-controlled studies and adopted an equivalent number of subjects in our studies. These authors also conducted a controlled sleep study with relatively smaller number of subjects. In our total sleep deprivation study, to compensate the small number, we set up 4 days of pre-deprivation phase and confirmed low variation of the skin yellowness (our primary endpoint) when they were under normal sleep. Given this result, we did not set a control group in the partial sleep deprivation study. However, we recognize the constraint from the small sample size. Respecting the importance of your comment, we would like to add a term of “preliminary” to the partial sleep deprivation study in the revised manuscript and state the necessity of larger population study to confirm our finding. We also describe the enrollment number (N=28 or N=10) in the relevant sentences for readers’ watch-out. We also expanded the constraint statements in the Discussion section. Below are the extraction of the revised parts:
Line 112 to Line123
“The recruitment of large number of healthy females who volunteer to have a sleep deprivation over their daily norms (e.g., schooling, working) is a common challenge in controlled sleep research. Kim et al. reported statistically significant changes in multiple skin biophysical parameters after one night of sleep deprivation with twenty-four healthy Korean women [13]. Sauver et al. examined the effect of acute sleep deprivation on vascular function in twelve healthy subjects [18]. Wessel et al. conducted a partial sleep deprivation study, 4 hours time in bed per night for 5 nights, with 13 healthy young men and concluded 5 nights of sleep restriction increased lymphocyte activation and the production of proinflammatory cytokines [19]. We have referred to the prior sleep deprivation protocols and recruited equivalent number of volunteers in our sleep deprivation studies.”
Line 136 to Line 137
We next conducted a preliminary study to elucidate the effects of repeated partial sleep deprivation.
Line 279 to Line 280
“We then conducted a preliminary study (N=10) to address whether repeated partial sleep deprivation aggravates facial skin yellowness.”
Line 354 to 358
“This is the first clinical study to assess the effects of sleep deprivation on facial skin yellowness. We recognize that there are several constraints in our studies. First, the number of subjects was not large due to a difficulty of recruiting volunteers in the longitudinal sleep deprivation protocols. Larger scale studies might be necessary to confirm our findings.”
Added subject number; Line 32, 33, 128, 138, 279.
The conclusion states that the increase in skin yellowness is probably a consequence of in situ events in the skin, which, however, is not supported by the results of the actual study. Although the authors stated the same as a shortcoming, it would be interesting to supplement the study with this data.
→ Thank you very much for your critical comment. We agree with your comment. According to your comment, we revised our conclusion as follows:
Line361 to 364
“In conclusion, both total sleep deprivation and repeated partial sleep deprivation enhance facial skin yellowness. In addition, increased yellowness and decreased hydration are likely to be differentially regulated in sleep deprivation.”
Thank you very much again for sharing your precious time to review this article. According to your comments, we revised the article.
We hope the revised article is now suitable for publication in JCM.

Reviewer 3 Report
Dear
1. There are several grammatical errors.
2. The number of cases is deficient; therefore, the results can't be reliable for citation in future research.
Author Response
Reply to the reviewer 3
- There are several grammatical errors.
→ Thank you very much for your helpful comment. According to your comment, the revised article was edited by the native English editor. The certification was included in the cover letter.
- The number of cases is deficient; therefore, the results can't be reliable for citation in future research.
→We appreciate your valuable comment. We totally agree with the criticality of this point. The recruitment of large number of healthy females who volunteer to have a sleep deprivation over their daily norms (e.g., schooling, working) is a common challenge in controlled sleep research although we understand that larger number of examinees is more adequate for solid conclusion. In particular, recruiting volunteers for the longitudinal and repeated partial sleep deprivation study was higher hurdle because the subject had to compromise their normal life for 12 days. When determining the number of subjects, we referred to the prior sleep-controlled studies and adopted an equivalent number of subjects in our studies. Those authors also conducted controlled sleep study with relatively smaller size. However, we recognize the constraint from the small sample size. Respecting the importance of this matter, we would like to add a term of “preliminary” to the partial sleep deprivation study in the revised manuscript and state the necessity of larger population study to confirm our finding. We also describe the enrollment number (N=28 or N=10) in the relevant sentences for readers watchout. We also expanded the constraint statements in the Discussion section. Below is the extraction of revised points:
Line 40 to Line 41
In conclusion, the facial skin yellowness is indeed increased by sleep deprivation in our preliminary clinical studies.
Line 112 to Line123
“The recruitment of large number of healthy females who volunteer to have a sleep deprivation over their daily norms (e.g., schooling, working) is a common challenge in controlled sleep research. Kim et al. reported statistically significant changes in multiple skin biophysical parameters after one night of sleep deprivation with twenty-four healthy Korean women [13]. Sauver et al. examined the effect of acute sleep deprivation on vascular function in twelve healthy subjects [18]. Wessel et al. conducted a partial sleep deprivation study, 4 hours time in bed per night for 5 nights, with 13 healthy young men and concluded 5 nights of sleep restriction increased lymphocyte activation and the production of proinflammatory cytokines [19]. We have referred to the prior sleep deprivation protocols and recruited equivalent number of volunteers in our sleep deprivation studies.”
Line 136 to Line 137
We next conducted a preliminary study to elucidate the effects of repeated partial sleep deprivation.
Line 279 to Line 280
“We then conducted a preliminary study (N=10) to address whether repeated partial sleep deprivation aggravates facial skin yellowness.”
Line 354 to 358
This is the first clinical study to assess the effects of sleep deprivation on facial skin yellowness. There are several constraints in our studies. The number of subjects was not large due to a difficulty of recruiting volunteers in the longitudinal sleep deprivation protocols. Larger scale studies might be necessary to confirm our findings.
Thank you very much again for sharing your precious time to review this article. According to your comments, we revised the article.
We hope the revised article is now suitable for publication in JCM.

Round 2
Reviewer 1 Report
I am happy with the responses addressed by the authors. I agree that the manuscript is fine to be accepted for publication.
Author Response
Reply to the Reviewer 1
I am happy with the responses addressed by the authors. I agree that the manuscript is fine to be accepted for publication.
→ Thank you very much for your kind consideration. We are pleased to hear your evaluation.

Reviewer 2 Report
The revision was made per the reviewer's recommendation, and I agree with the article's publication in the revised form.
Author Response
Reply to the Reviewer 2
The revision was made per the reviewer's recommendation, and I agree with the article's publication in the revised form.
→ Thank you very much for your kind consideration. We are pleased to hear your evaluation.

Reviewer 3 Report
Dear
The main issue is "the number of cases is deficient".
Author Response
Reply to the Reviewer 3
The main issue is "the number of cases is deficient".
→ Thank you very much for your critical review. We understand your criticism. The number of subjects in the previous sleep deprivation studies were 12 (Ref. 18), 13 (Ref. 19) and 24 (Ref. 13). In the present report, we enrolled 28 subjects in the total sleep deprivation study and 10 subjects in the preliminary partial sleep deprivation study. As you pointed out, the number of subjects may be small. Therefore, we explained the subject number in the Materials and Methods (line 114-125). We also mentioned the small number limitation in the Discussion (line 360-364) as follows;
Line 114-125
“The recruitment of large number of healthy females who volunteer to have a sleep deprivation over their daily norms (e.g., schooling, working) is a common challenge in controlled sleep research. Kim et al. reported statistically significant changes in multiple skin biophysical parameters after one night of sleep deprivation with twenty-four healthy Korean women [13]. Sauver et al. examined the effect of acute sleep deprivation on vascular function in twelve healthy subjects [18]. Wessel et al. conducted a partial sleep deprivation study, 4 h time in bed per night for 5 nights, with 13 healthy young men and concluded 5 nights of sleep restriction increased lymphocyte activation and the production of proinflammatory cytokines[19]. We have referred to the prior sleep deprivation protocols and recruited equivalent number of volunteers in our sleep deprivation studies.”
Line 360-364
“This is the first clinical study to assess the effects of sleep deprivation on facial skin yellowness. We recognize that there are several constraints in our studies. First, the number of subjects was not large due to a difficulty of recruiting volunteers in the longitudinal sleep deprivation protocols. Larger scale studies might be necessary to confirm our findings.”
Thank you very much again for reviewing our revised article. We appreciate your kind consideration and understanding about the number of subjects.
We hope the re-revised article is now suitable for publication.
